

# Examining the stress-burnout relationship: the mediating role of negative thoughts

Ko-Hsin Chang[1,*], Frank J.H. Lu[2,*], Theresa Chyi[3], Ya-Wen Hsu[4], Shi-Wei Chan[5] and Erica T.W. Wang[6]

[1] Department of Physical Education, Chinese Culture University, Taipei City, Taiwan
[2] Graduate Institute of Sport Coaching Science, Chinese Culture University, Taipei City, Taiwan
[3] Department of Exercise and Health Promotion, Chinese Culture University, Taipei City, Taiwan
[4] Department of Physical Education, Health, and Recreation, National Chiayi University, Chia-Yi, Taiwan
[5] Graduate Institute of Physical Education, National Taiwan Sport University, Kweishan, Taoyuan, Taiwan
[6] Office of Physical Education and Sports Affairs, Feng Chia University, Taichung, Taiwan
[*] These authors contributed equally to this work.

## ABSTRACT

**Background**. Using *Smith*'s (*1986*) cognitive-affective model of athletic burnout as a guiding framework, the purpose of this study was to examine the relationships among athletes' stress in life, negative thoughts, and the mediating role of negative thoughts on the stress-burnout relationship.

**Methods**. A total of 300 college student-athletes (males = 174; females = 126, $M_{age}$ = 20.43 y, $SD$ = 1.68) completed the College Student Athlete's Life Stress Scale (CSALSS; *Lu et al., 2012*), the Automatic Thoughts Questionnaire (ATQ; *Hollon & Kendall, 1980*), and the Athlete Burnout Questionnaire (ABQ; *Raedeke & Smith, 2001*).

**Results**. Correlational analyses found that two types of life stress and four types of negative thoughts correlated with burnout. Additionally, hierarchical regression analyses found that four types of negative thoughts partially mediated the stress-burnout relationship.

**Discussion**. We concluded that an athlete's negative thoughts play a pivotal role in predicting athletes' stress-burnout relationship. Future study may examine how irrational cognition influences athletes' motivation and psychological well-being.

## INTRODUCTION

Being successful in sports brings fame and monetary rewards to athletes, but the costs of athletic success are high. To achieve this, athletes not only have to engage in sports at a very young age but also need to participate in intensive training every day and compete year-round (*Gould & Whitley, 2009*). Research has suggested that intensive involvement in sports is not beneficial. One of the potential negative outcomes is burnout. Burnout is a complex psycho-physiological symptom characterized by physical and emotional exhaustion, de-evaluation in the sport, and reduced sense of accomplishment (*Raedeke & Smith, 2004*). Although the prevalence of burnout has been minimal in both female athletes (1%–9%) and male athletes (2%–6%) (*Gustafsson et al., 2007*), studies have indicated that

Corresponding author
Frank J.H. Lu,
ljh25@ulive.pccu.edu.tw,
frankjlu@gmail.com

the burnout experience is detrimental to an athlete's psychological well-being (*Franciscoa, Vílcheza & Valesc, 2016*; *Gould et al., 1997*), resulting in depression (*Franciscoa, Vílcheza & Valesc, 2016*), increasing interpersonal difficulties (*Smith, 1986*), and triggering drop-out (*Gould & Whitley, 2009*).

To explain the process of athlete burnout, *Smith (1986)* proposed a "cognitive-affective model of athletic burnout," which contends that burnout is a reaction to chronic stress and it comprises situational, cognitive, physiological, and behavioral components that progress in four predicted stages. The first stage is triggered by the athletes' perceived situational demands (e.g., high conflicting demands, overload training, parental expectations, or coaches' pressures). The second stage involves cognitive appraisal to interpret these demands. In particular, athletes at this stage will appraise the balance between challenges and resources with potential consequences. When athletes perceive that demands surpass personal resources and consequences will be severe, the process will move to the third stage, which involves physiological and psychological responses, such as anxiety, tension, insomnia, and illness. Finally, physiological and psychological responses lead to burnout (i.e., rigid and inappropriate behavior, decreased performance, and withdrawal from the activity).

Based on the cognitive-affective model of athletic burnout, many studies attempt to examine the association between an athlete's stress and burnout; and the mechanism underlying the stress-burnout relationship. For example, *Cresswell & Eklund (2006)* interviewed 15 New Zealand professional rugby players and they found that these athletes had many negative experiences, such as injury, competitions, performance demands, anti-rest culture, media pressure, and training loadings associated with burnout. In addition, *Gustafsson et al. (2008)* interviewed 11 Swedish athletes and found that multiple demands, such as "too much sport," "lack of recovery," and "high expectations," were considered primary causes of burnout. Recently, Francisco and colleagues (*2016*) investigated 453 Spanish athletes and found that perceived stress accounted for 43% of the variance for burnout. Furthermore, perceived stress and burnout jointly accounted for 50% of the variance in depression.

In addition, researchers have also examined the mechanisms underlying the stress-burnout relationship. For example, *Gustafsson & Skoog (2012)* measured 217 participants for their level of optimism, perceived stress, and burnout. They found perfectionism, perceived stress, and burnout all correlated, and perceived stress fully mediated the optimism-burnout relationship. Furthermore, when considering that hope is a positive buffer for stress-related burnout, *Gustafsson et al. (2013)* investigated 238 Swedish youth soccer players and examined the relationships among hope, perceived stress, and burnout. They found that hope negatively correlated with perceived stress and all three dimensions of burnout. Additionally, the relationship between hope and burnout was fully mediated by stress and positive effects. Recently, *Lu et al. (2016)* adopted an interactive perspective to examine the conjunctive effects of athletes' resilience and coaches' social support in moderating the stress-burnout relationship. Results indicated that athletes' resilience correlated with the level of the coaches' social support, and this effect moderated athletes' stress-burnout relationship. Specifically, athletes' resilience interacted with coaches'

informational/tangible social support in moderating athletes' stress-burnout relationship in both high and low life stress situations.

Despite these efforts, it is unknown what role negative thoughts play in the stress-burnout relationship. The study of negative thoughts can be traced back to *Beck*'s (*1976*) works on depressed patients. In his works, he proposed there are three key elements: negative views about the world, negative views about the future, and view of the self, formed a negative cognition triad model. *Beck (1976)* contended that some individuals encounter major life events and become depressed are because of cognitive distortion and irrational thoughts. In non-sports settings, it was found that negative thoughts are not only linked to depression (*Clark & Goosen, 2009*), but they are also associated with pain/psychological distress (*Gil et al., 1990*), fatigue (*Arpin-Cribbie & Cribbie, 2007*), and a dysphoric mood (*Stiles & Gotestam, 1989*).

Negative thoughts play a pivotal role in the cognitive-affective model of athletic burnout. Specifically, Smith contends that when encountering environmental stressors, athletes engage in a cognitive appraisal to evaluate whether these stressors are threatening or whether they can handle them (*Smith, 1986*). Thus, the ways in which athletes appraise the stressors, both from positive and negative perspectives, are worthy of investigation.

In sports, there are several types of negative thoughts. The first example is negative self-talk. Self-talk is a cognitive strategy that athletes use to direct their attention or psych themselves up (*Hatzigeorgiadis et al., 2011*). Literature has indicated that self-talk can be either negative or positive, and it can influence athletes' affect, cognition, and performance (*Chang et al., 2014*; *Zourbanos et al., 2010*).

The second example is the negative expectation. Negative expectation refers to one's expectation that something positive will not happen in the future. In sports, the negative expectation is closely related to cognitive anxiety. For example, Martens and colleagues (*1990*) conducted a series of studies to develop a state anxiety measure termed the "Competitive State Anxiety Inventory-2" (CSAI-2). The CSAI-2 consists of three factors: somatic anxiety, cognitive anxiety, and confidence. Cognitive anxiety consists of negative expectations (e.g., I am concerned I may not do as well in this competition as I could). Empirical studies found that negative expectations associated with anxiety and lowered overall performance (*Suhr & Gunstad, 2004*). Some other research also found negative expectations were associated with poor sports injury recovery (*Arvinen-Barrow et al., 2016*).

The third example is self-defeating attribution. Self-defeating refers to attribute success as external, unstable, or uncontrollable variables (e.g., others' help, luck, weather), while attribute failure refers to internal, stable, and controllable reasons (e.g., ability, task, referee.) Athletes use different explanations for their achievements and social events (*Allen, 2012*), such as explaining failures and successes (e.g., *Gernigon & Delloye, 2003*), interpersonal interactions (e.g., *Dimmock, Grove & Eklund, 2005*), and injuries (e.g., *Brewer, 1999*). Though self-defeating behavior is not often seen in males, it is frequently found in female athletes (e.g., *Green & Holeman, 2004*). Research indicated that athletes' causal attributions influence their motivation, affect, and performance (*Allen , 2012*).

Using the aforementioned theoretical and empirical background, the purpose of this study was to examine the relationships among athletes' life stressors (both in sports and in

general life), negative thoughts (i.e., personal maladjustment, the desire for change, negative sense of self, giving up, learned helplessness, and negative expectations), and burnout. We also examined the mediating role of negative thoughts on the life stress-burnout relationship. Because past research has indicated that negative life events precede negative thoughts (e.g., *Lakey & Tanner, 2013*), we hypothesized that athletes' life stress would positively correlate with negative thoughts. Furthermore, because research has found that negative thoughts are associated with maladaptive outcomes (e.g., *Arpin-Cribbie & Cribbie, 2007*; *Clark & Goosen, 2009*; *Gil et al., 1990*; *Stiles & Gotestam, 1989*), we hypothesized that negative thoughts would positively correlate with athlete burnout. Moreover, because negative thoughts are likely related to athletes' life stress and burnout, it was hypothesized that negative thoughts would play a mediating role in an athlete's life stress-burnout relationship. The rationale was based on *Barron & Kenny* (*1986*, p. 1176), who suggested that a mediator is a third variable that explains how external physical events have internal psychological significance.

## MATERIALS & METHODS

### Participants

Through a convenience sampling, 300 student-athletes (males = 174; females = 126, $M_{age} = 20.43$ y, $SD = 1.68$) from five colleges voluntarily participated in this study. During data collection, all participants were engaged in their regular training seasons. All participants had been participating in either team sports (basketball, softball, volleyball) or individual sports (gymnastics, air pistol, tennis, track and field, table tennis, tug of war, archery, or golf) for an average of 9.83 y ($SD = 3.42$ y) of training and competition experiences.

### Measures

#### General characteristics

We used a demographic questionnaire to collect information about participants' age, gender, types of sports, training hours per day, training frequency per week, and years of athletic experience.

#### Life stressors

To measure participants' life stressors, we used the 24-item College Student-Athletes Life Stress Scale (CSALSS; *Lu et al., 2012*) to evaluate situations that athletes encounter in their daily life and in sports, which represent major stressors. These stressors all had two major components: sport-specific stress (i.e., coach relationships, performance demand, sports injuries, and training adaptation) and general life stress (i.e., family relationships, interpersonal relationships, romantic relationships, academic requirements). For instance, on various questions, such as "I am annoyed with my coach's bias against me" or "I am annoyed by my repetitive injury," participants responded about the frequency of the event on a six-point Likert scale ranging from 1 (*never*) to 6 (*always*). The Cronbach's $\alpha$ for these factors in the present study ranged from 0.69 to 0.87, indicating that the measure was reliable. Lu and colleagues (*2012*) reported CSALSS with adequate concurrent and

discriminant validity. Also, related research indicated that CSALSS correlated with global perceived stress (*Chiu et al., 2016*) which provides evidence of predictive validity.

### Negative thoughts

A 30-item Automatic Thoughts Questionnaire (ATQ; *Hollon & Kendall, 1980*) was applied to assess the frequency of occurrence for 30 types of thoughts in the previous week. Participants rated each single negative self-statement on a 5-point Likert scale ranging from 1 (*not at all*) to 5 (*always*). The ATQ was divided into four subscales: (a) Personal Maladjustment and Desire for Change (PMADC) (e.g., I wish I were a better person); (b) Negative Self-Concept (NSC) (e.g., I am a failure); (c) Giving Up/Helplessness (GU/H) (e.g., I cannot finish anything); and (d) Negative Expectations (NE) (e.g., I cannot get things together). All items represented experiences and thoughts of depression. Total scores for ATQ ranged from 30 (no or a minor depression) to 150 (major depression). *Hollon & Kendall (1980)* reported ATQ can differentiate depressed and non-depressed individuals. Also, ATQ has been validated in Norwegian and found positively correlated with Beck Depression Inventory (*Chioqueta & Stiles, 2004*).

### Burnout

An 11-item Athlete Burnout Questionnaire (ABQ; *Lu, Chen & Cho, 2006*) revised from *Raedeke & Smith (2001)* burnout scale was used to assess athletes' burnout. The ABQ was divided into three subscales: emotional or physical exhaustion, sports devaluation, and a reduced sense of accomplishment. Sample questions assessing emotional or physical exhaustion included "I feel so tired from my training that I have trouble finding energy to do anything else." To assay for a reduced sense of accomplishment, questions included "I am accomplishing many worthwhile things in sports." Finally, to assess sports devaluation, questions included "The effort I spend in sports would be better spent doing other things." Among all the items on the ABQ, item 1 and 11 were reversed to score. To complete the ABQ, participants chose options on a five-point Likert scale ranging from 1 (*almost never*) to 5 (*always*) to assess their experiences. The higher the number participants identified, the higher the degree of burnout. The results were internally consistent with scores of 0.85, 0.86, and 0.63. In this research, we used a composite score by adding the three subscales together. Lu and colleagues (*2006*) found ABQ positively correlated with amotivation and negatively correlated with intrinsic motivation which provides concurrent validity and discriminant validity.

## Procedure

After receiving approval from a local institutional review board (TSMH IRB No. 16-094-B1), the research team contacted the coach or administrator of a target team and asked permission to invite his/her team as participants. After accepting, we visited the target team one hour before they finished their regular training. Before administering the questionnaire, we explained the general purpose of the study, the methods to complete questionnaires, and the rights associated with being a participant. To prevent social desirability effects, we informed participants that this was a study to explore college students' life experiences, and there were no right or wrong answers. Additionally, we asked participants to answer

the questions as truthfully as possible, and all responses would be confidential. After the briefing, participants who were interested in participating in this study signed a consent form and completed the investigation package. It took about 15 min to complete the questionnaires.

## Statistical analyses
### Preliminary analyses

First, a descriptive statistical analysis was applied to examine the properties of the collected data, including skewness, kurtosis, means, standard deviations, outliers, and missing data for all variables. The skewness (0.10–1.38) and kurtosis ($-0.10-1.38$) were found to be in an acceptable range. Gender, age, and competition levels for all variables were examined using a $t$-test and analysis of variance (ANOVA). The results indicated that there was a significant gender difference in the subscales of "personal maladjustment and desire for change (PMADC)" and "giving up/helplessness (GU/H)" for the ATQ. Therefore, in the subsequent hierarchical regression analyses, gender was controlled for when examining the mediating effects of these two factors.

### Hypothesis testing

A Pearson product-moment correlation analysis was used to examine the relationships of all variables and provide information about whether the predictor variable (i.e., two types of life stress), criterion variable (i.e., burnout), and mediating variables (i.e., four types of negative thoughts) correlated. This analysis was conducted as a prerequisite analysis for testing mediator effects (*Barron & Kenny, 1986*, p. 1174). According to Barron and Kenny's (*1986*) suggestions, we examined mediation by assuming that the following conditions were met: (a) two types of life stress should predict four types of negative thoughts, (b) four types of negative thoughts should predict burnout, and (c) two types of life stress should predict burnout. If all three conditions were met, the subsequent mediating effects of four types of negative thoughts on the relationship between two types of life stress and burnout were further analyzed.

A series of hierarchical regression analyses were applied to examine the main effects and interaction effects. First, two types of life stress (general life stress and sport-specific stress) were entered into the regression at the first step. Then, four types of negative thoughts were entered at the second step. Two types of life stress and four types of negative thoughts were simultaneously entered at the third step. The final test for mediation was to check whether two types of life stress predicted burnout when four types of negative thoughts were controlled for.

## RESULTS

### Descriptive statistics and correlations matrix

Table 1 shows that all measures had appropriate internal reliability (ranging from 0.76–96). These results indicated that athlete burnout positively correlated with a total score for negative thoughts, four types of negative thoughts, and two types of life stress. Among these correlations, burnout correlated more significantly with two types of life stress than

**Table 1  Correlation matrix and descriptive statistics for study variables.**

|  | ABQ | ATQ | PMADC | NSC | GU/H | NE | Sport stress | General stress |
|---|---|---|---|---|---|---|---|---|
| 1. ABQ | 1.00 | .47** | .46** | .47** | .41** | .42** | .54** | .49** |
| 2. ATQ |  | 1.00 | .86** | .88** | .96** | .95** | .49** | .52** |
| 3. PMADC |  |  | 1.00 | .79** | .74** | .69** | .53** | .56** |
| 4. NSC |  |  |  | 1.00 | .79** | .80** | .50** | .53** |
| 5. GU/H |  |  |  |  | 1.00 | .92** | .44** | .44** |
| 6. NE |  |  |  |  |  | 1.00 | .40** | .44** |
| 7. Sport Stress |  |  |  |  |  |  | 1.00 | .68** |
| 8. Gen. Stress |  |  |  |  |  |  |  | 1.00 |
| *Mean* | 2.48 | 1.86 | 2.36 | 1.92 | 1.76 | 1.62 | 2.61 | 2.43 |
| *SD* | 0.76 | 0.65 | 0.77 | 0.73 | 0.70 | 0.67 | 0.85 | 0.96 |
| $\alpha$ | .89 | .96 | .84 | .76 | .91 | .94 | .89 | .90 |

**Notes.**

ABQ, athlete burnout; ATQ, total score of automatic negative thoughts; PMADC, personal maladjustment and desire for change; NSC, negative self-concept; GU/H, giving up/helplessness; NE, negative expectations; Sport Stress, sport-specific life stress; Gen. Stress, general-life stress.

** $p < .01$.

negative thoughts. Four types of negative thoughts positively correlated with each other, as did two types of life stress.

## Predictions for the effects of life stress and negative thoughts on burnout

Before performing mediating analyses, we examined the relationships among life stress, negative thoughts, and burnout. As Table 2 shows, sports-specific life stress predicted four types of negative thoughts ($\beta = 0.53$; $\beta = 0.50$; $\beta = 0.44$; $\beta = 0.40$, $p < 0.01$) in Model 1. In Model 2, both sports-specific life stress ($\beta = 0.54$; $\beta = 0.54$; $\beta = 0.54$ ; $\beta = 0.54$, $p < 0.01$) and four types of negative thoughts ($\beta = 0.46$; $\beta = 0.47$; $\beta = 0.41$; $\beta = 0.42$, $p < 0.01$) predicted burnout. Thus, these results demonstrated that it these data can be further analyzed for mediating effects. Also, as Table 3 illustrates, general life stress predicted four types of negative thoughts ($\beta = 0.56$; $\beta = 0.53$; $\beta = 0.44$; $\beta = 0.44$, $p < 0.01$) in Model 1. In Model 2, both general life stress ($\beta = 0.49$; $\beta = 0.22449$; $\beta = 0.49$; $\beta = 0.49$, $p < 0.01$) and four types of negative thoughts ($\beta = 0.46$; $\beta = 0.47$; $\beta = 0.41$; $\beta = 0.42$, $p < 0.01$) predicted burnout. Thus, these results indicated that these data can be further analyzed for mediating effects.

## Mediating effects of negative thoughts on the sports-specific life stress-burnout relationship

The upper part of Table 4 shows the mediating effects of four types of negative thoughts on the sports-specific life stress-burnout relationship. The first block illustrates that personal maladjustment and the desire for change (PMADC) mediated the sports-specific life stress-burnout relationship ($\beta = 0.54$ changed to $\beta = 0.41$). The second block demonstrates that negative self-concept (NSC) mediated the sports-specific life stress-burnout relationship ($\beta = 0.54$ changed to $\beta = 0.41$). The third block demonstrates that giving up and

**Table 2  Simple regression of sport- life stress and negative thoughts on burnout.**

| Variables | PMADC | | NSC | | GU/H | | NE | |
|---|---|---|---|---|---|---|---|---|
| | $\beta$ | $\Delta R^2$ | $\beta$ | $\Delta R^2$ | $\beta$ | $\Delta R^2$ | $\beta$ | $\Delta R^2$ |
| Regression 1 | | | | | | | | |
| Sport stress | .53** | .28** | .50** | .25** | .44** | .19** | .40** | .16** |
| Regression 2 | | | | | | | | |
| Sport stress | .54** | .29** | .54** | .29** | .54** | .29** | .54** | .29** |
| Negative Thoughts | .46** | .21** | .47** | .22** | .41** | .16** | .42** | .18** |

Notes.
The dependent variables of regression 1 was negative thoughts, and regression 2 was burnout. All abbreviations were as indicated in Table 1.
** $p < .01$.

**Table 3  Simple regression of general-life stress and negative thoughts on burnout.**

| Variables | PMADC | | NSC | | GU/H | | NE | |
|---|---|---|---|---|---|---|---|---|
| | $\beta$ | $\Delta R^2$ | $\beta$ | $\Delta R^2$ | $\beta$ | $\Delta R^2$ | $\beta$ | $\Delta R^2$ |
| Regression 1 | | | | | | | | |
| Gen-life stress | .56** | .31** | .53** | .27** | .44** | .19** | .44** | .19** |
| Regression 2 | | | | | | | | |
| Gen-life stress | .49** | .24** | .49** | .24** | .49** | .24** | .49** | .24** |
| Negative Thoughts | .46** | .21** | .47** | .22** | .41** | .16** | .42** | .18** |

Notes.
The dependent variables of regression 1 was negative thoughts, and regression 2 was burnout.
All abbreviations were as indicated in Table 1.
** $p < .01$.

helplessness (GU/H) mediated the sports-specific life stress-burnout relationship ($\beta = 0.54$ changed to $\beta = 0.45$). The fourth block shows that negative self-concept (NE) mediated the sports-specific life stress-burnout relationship ($\beta = 0.54$ changed to $\beta = 0.44$). Sobel tests of PMADC, NSC, GU/H, and NE were 6.86, 6.74, 5.63, and 5.81 ($z > 1.96$ and $p < 0.05$), respectively.

## Mediating effects of negative thoughts on the general life stress-burnout relationship

The lower part of Table 4 shows the mediating effects of four types of negative thoughts on the general life stress-burnout relationship. The first block illustrates that personal maladjustment and the desire for change (PMADC) mediated the general life stress-burnout relationship ($\beta = 0.49$ changed to $\beta = 0.34$). The second block demonstrates that negative self-concept (NSC) mediated the sports-specific life stress-burnout relationship ($\beta = 0.49$ changed to $\beta = 0.34$). The third block shows that giving up and helplessness (GU/H) mediated the general life stress-burnout relationship ($\beta = 0.49$ changed to $\beta = 0.39$). The fourth block shows that negative self-concept (NE) mediated the general life stress-burnout relationship ($\beta = 0.49$ changed to $\beta = 0.38$). Sobel tests for PMADC, NSC, GU/H, and NE p were 7.03, 7.02, 5.68, and 5.87 ($z > 1.96$ and $P < 0.05$), respectively.

**Table 4  Mediating models of negative thoughts on life stress-burnout relationships.**

| Variables | PMADC Model 1 β | PMADC Model 1 $\Delta R^2$ | PMADC Model 2 β | PMADC Model 2 $\Delta R^2$ | NSC Model 1 β | NSC Model 1 $\Delta R^2$ | NSC Model 2 β | NSC Model 2 $\Delta R^2$ | GU/H Model 1 β | GU/H Model 1 $\Delta R^2$ | GU/H Model 2 β | GU/H Model 2 $\Delta R^2$ | NE Model 1 β | NE Model 1 $\Delta R^2$ | NE Model 2 β | NE Model 2 $\Delta R^2$ |
|---|---|---|---|---|---|---|---|---|---|---|---|---|---|---|---|---|
| Step 1 | | .29** | | | | .29** | | | | .29** | | | | .29** | | |
| Sport stress | .54** | | .41** | | .54** | | .41** | | .54** | | .45** | | .54** | | .44** | |
| Step 2 | | | | .33** | | | | .34** | | | | .32** | | | | .34** |
| PMADC | | | .24** | | | | | | | | | | | | | |
| NSC | | | | | | | .26** | | | | | | | | | |
| GU/H | | | | | | | | | | | .21** | | | | | |
| NE | | | | | | | | | | | | | | | .25** | |
| Step 1 | | .24** | | | | .24** | | | | .24** | | | | .24** | | |
| General stress | .49** | | .34** | | .49** | | .34** | | .49** | | .39** | | .49** | | .38** | |
| Step 2 | | | | .28** | | | | .30** | | | | .28** | | | | .29** |
| PMADC | | | .27** | | | | | | | | | | | | | |
| NSC | | | | | | | .29** | | | | | | | | | |
| GU/H | | | | | | | | | | | .24** | | | | | |
| NE | | | | | | | | | | | | | | | .26** | |

**Notes.**

The upper part was the mediation analysis of negative thoughts on sport-specific life stress-burnout relationship.
Sobel tests of PMADC, NSC, GU/H and NE were 6.86, 6.74, 5.63 and 5.81 ($>1.96$ and $p < .05$), respectively.
The lower part was the mediation analysis of negative thoughts on general-life stress-burnout relationship.
Sobel tests of PMADC, NSC, GU/H and NE p were 7.03, 7.02, 5.68 and 5.87 ($z > 1.96$ and $p < .05$), respectively.
All abbreviations were as indicated in Table 1.
** $p < .01$.

## DISCUSSION

To determine whether negative thoughts might influence the stress-burnout relationship for athletes, this study examined the relationships among athletes' life stress, negative thoughts, and burnout; and the potential mediating effects of negative thoughts on the life stress-burnout relationship. Results indicated that two types of athletes' life stress, four types of negative thoughts, and burnout were all positively correlated. Also, four types of negative thoughts mediated a general life- and sports-specific stress-burnout relationship.

Our results provide several theoretical implications for researchers. By incorporating negative thoughts into *Smith*'s (*1986*) cognitive-affective model of athletic burnout, our study advances our knowledge of how negative thoughts underlie the life stress-burnout relationship for athletes. We found that the association between athletes' life stress and burnout was partially explained by negative thoughts. Past research examining the mechanisms underlying the stress-burnout relationship generally found that positive variables, such as self-efficacy (e.g., *Schwarzer & Hallum, 2008*), optimism (*Gustafsson & Skoog, 2012*), teaching strategy (e.g., *Ben-Ari, Krole & Har-Even, 2003*), hardiness (e.g., *Chan, 2003*), coping and social support (e.g., *Koeske & Koeske, 1989*; *Raedeke & Smith, 2004*), or perceived accomplishments (e.g., *Koeske & Koeske, 1989*), mediate or moderate the stress-burnout relationship. Our study provides new evidence that negative thoughts can change the stress-burnout relationship. These results should prompt future research to examine how positive and negative variables influence the stress-burnout relationship.

The mediating effect of negative self-concept (NSC) on the two types of life stress-burnout relationships suggests that there is a role for the self-concept in predicting athletes' motivation, affect, and behavior (*Gallia & Gonzalez, 2014*). Past research found that when athletes had low self-esteem, they either adopted self-handicapping strategies in sports (*Finez & Sherman, 2012*) or were susceptible to developing eating disorders (*McLester, Hardin & Hoppe, 2014*). Thus, it is important to monitor athletes' self-related thought patterns. Additionally, coaches, parents, and sports professionals need to build a positive motivational culture in which athletes' self-concept will not be threatened in losing situations (*Smith, Smoll & Cumming, 2007*).

The effect of giving up and helplessness (GU/H) on the relationship between two types of life stress and burnout prompted us to examine athletes' causal explanation about their failure in sports. Why one gives up and displays helplessness derives from attributing failure to internal and stable reasons (*Weiner, 1972*). As previously stated, female athletes tend to attribute success to external, unstable, and uncontrollable causes, while attributing failure to internal, stable, and controllable reasons (*Green & Holeman, 2004*). Therefore, how to change athletes' thought patterns and attribution retraining for those with self-defeating attribution is key for coaches and sport psychology consultants.

The mediating effects of negative expectations on the relationship between two types of life stress and burnout imply that when athletes tend to be negative about their future, they will increase the life stress-burnout relationship. Although no specific investigation focused on the sources of negative expectations in sports, it has been shown that social agents, such as coaches, parents, peers, and teachers, play a significant role in formulating

athletes' expectations (*Weinberg & Gould, 2015*). Also, *Bandura*'s (*1997*) self-efficacy theory indicated that personal accomplishments, vicarious experiences, verbal persuasion, and physiological responses are the sources of efficacy expectations. Thus, developing strategies to facilitate athletes' positive expectations and reduce negative expectations will be effective for reducing athlete burnout.

The mediating effect of personal maladjustment and the desire for change (PMADC) on the life stress-burnout relationship suggests that such depressogenic thoughts may increase athletes' burnout. As previously stated, athletes encounter many stressors from either life or sports. Literature has suggested that chronic stress leads to depression (*Risch et al., 2009*), increasing hopelessness and suicidal ideation (*Glick et al., 2012*). Furthermore, it has been found that depressed adolescent athletes are susceptible to doping (*Blank et al., 2016*). Therefore, future studies should examine how athletes' personal maladjustment and desire for change influence unhealthy behaviors, motivations, and overall well-being.

In addition, the sources of athlete burnout are worthy of discussion. The extant sport burnout literature suggests that environmental factors (e.g., excessive training, stressful social relationship, negative performance demands, low social support), personal factors (e.g., perfectionism, trait anxiety, low autonomy), and motivational factors (e.g., ego orientation, performance motivational climate) are causes of athletic burnout (*Gould & Whitley, 2009*; *Gustafsson, Kentta & Hassmen, 2011*). Thus, our results add to this body of knowledge by showing that athletes' thought patterns can be associated with burnout. Specifically, *Smith*'s (*1986*) cognitive-affective model of athletic burnout emphasizes the role of the cognitive process in the stress-burnout relationship. Therefore, how another type of thought, such as positive thoughts (*Bryant & Baxter, 1997*), influences an athlete's life stress-burnout relationship is worthy of investigation in the future. The effects of positive thoughts (e.g., positive daily functioning, positive self-evaluation, positive evaluation of self, and positive future expectation) can be new directions for athlete burnout research.

The association between life stress and negative thoughts is also worthy of discussion. In the past, research on negative thoughts found that neuroticism (*Kercher, Rapee & Schniering, 2009*), stress management skills (*Clark & Goosen, 2009*), social influence (*Lakey & Tanner, 2013*), and life events (*Kercher, Rapee & Schniering, 2009*) are sources of irrational thoughts. Our results advance this body of knowledge by showing that athletes' life stress, regardless of their general-life stress or sport-specific life stress, is associated with negative thoughts. In addition to life stress, research has indicated that there are many types of life stressors, such as work and family (*Bacharach, Bamberger & Conley, 1991*), organizational stress (*Arnold, Fletcher & Daniels, 2013*), and both general and sports-specific life stressors (*Lu et al., 2012*). Hence, how different types of life stressors interact with negative thoughts to influence an individual's health and well-being is another direction for future research.

## Limitations and future suggestions

There are several limitations that should be addressed here. First, our study is cross-sectional in nature. Therefore, the causes and effects of the stress-burnout relationship and the mediating effect of negative thoughts on the stress-burnout relationship are not

warranted. We suggest that future studies adopt a longitudinal approach to observe how life stress associated with negative thoughts subsequently predicts burnout (*Eklund & DeFreese, 2015*). Second, these data were collected from Taiwanese student-athletes. Thus, our results may not be generalizable to different cultures. We recommend that researchers adopt a similar approach for examining the relationships among life stressors, negative thoughts, and burnout in a different culture. Furthermore, our sample consisted of Division I college athletes. Whether our results could be generalized to other athletes, such as professional athletes or junior athletes, needs to be examined. Moreover, since we adopted *Smith*'s (*1986*) cognitive-affective model of athletic burnout as a guiding framework for our study, it is possible that the three sets of variables (life stress, negative thoughts, and burnout) could be causally related in several ways. For example, life stress could precipitate burnout, which could lead to negative thoughts; and burnout plays a mediating role in the model. We suggest that future studies examine these potential models by using related theoretical frameworks. Last, we used hierarchical regression to analyze the mediating effects of negative thoughts on life stress-burnout relationship, it is suggested that other statistical techniques, such as structural equation modeling (SEM), path analysis and factor analysis, can be applied to examine the causal effects of the model depends on the theories and aims of the study (*Musil, Jones & Warner, 1998*).

## CONCLUSIONS

Burnout is a significant issue for athletes, administrators, parents, coaches, and sport psychology consultants because it can be detrimental to a youth's psychosocial development and well-being. Athletes may have different sources of burnout. Our study found that both life stress and negative thoughts interact to influence athletes' burnout. We consider our study to be a starting point for researchers to examine the role of negative thoughts in athlete burnout. We suggest sports professionals should be aware of athletes' burnout and build a healthy competitive environment for youth athletes.

### Funding
The authors received no funding for this work.

### Competing Interests
The authors declare there are no competing interests.

### Author Contributions
- Ko-Hsin Chang conceived and designed the experiments, performed the experiments, wrote the paper.
- Frank J.H. Lu conceived and designed the experiments, wrote the paper, reviewed drafts of the paper, monitoring the whole process of the study.
- Theresa Chyi conceived and designed the experiments, performed the experiments, wrote the paper, prepared figures and/or tables.

- Ya-Wen Hsu performed the experiments, analyzed the data, wrote the paper, prepared figures and/or tables.
- Shi-Wei Chan and Erica T.W. Wang analyzed the data, contributed reagents/materials/analysis tools.

## Human Ethics

The following information was supplied relating to ethical approvals (i.e., approving body and any reference numbers):

The Antai Medical Care Cooperation Antai-Tian-Sheng Memorial Hospital Institutional Review Board granted ethical approval to carry out this study.

## Data Availability

The raw data has been provided as a Supplemental File.

## Supplemental Information

Supplemental information for this article can be found online at http://dx.doi.org/10.7717/peerj.4181#supplemental-information.

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
