# Peer review of "Examining the stress-burnout relationship: the mediating role of negative thoughts"

_PeerJ, doi:10.7717/peerj.4181_

## Round 0.1 · original submission · Major Revisions

I now have received two reviewers' comments. Although both reviewers expressed their interest in your study, several aspects of this manuscript should be revised to improve its clarity. Their observations are presented with clarity so I'll not risk confusing matters by belaboring or reiterating their comments. While I might quibble with the occasional point, I note that I regard the reviewers' opinions as substantive and well-informed. I believe that all of the highlighted reservations require contemplation and appropriate attention in revising the document if it is to contribute appropriately to Peer J and the extant literature. Please revise or refute according to the two reviewers' comments and provide a point by point reply in addition to the revised manuscript.

In addition, both reviewers also pointed out the language issue that dramatically impaired the quality of your manuscript. Therefore, I'd suggest you to have your revised manuscript gone through a thorough language editing by a professional native speaking editor before your resubmission.

Tsung-Min Hung, Ph.D.
PeerJ editor
Distinguished professor
Department of Physical Education
National Taiwan Normal University

Reviewer 1 ·

Basic reporting

1. Although the writing is generally clear, the manuscript could benefit greatly from a thorough proofreading (and modification where necessary) by a native English-speaker.

2. The nature of the interaction and conjunctive moderation noted on lines 124-125 should be described. Similarly, the nature of the mediation and moderation mentioned on l. 147-154 should be described.

3. On line 132, it should be "negative cognitive triad."

Experimental design

1. Evidence in support of the validity of the CSALSS, ATQ, and ABQ should be summarized in the Results section.

Validity of the findings

1. Although the cross-sectional nature of the data are duly noted in the Discussion section on lines 430-434, language that implies that causal relations were investigated and established is used throughout the manuscript (e.g., "influencing" on line 67, "source of burnout" on line 411, "sources of negative thoughts" on line 423). Such language should be removed from the manuscript.

2. Similarly, the process described on lines 89 to 97 cannot be examined with cross-sectional data. In a cross-sectional design, the terms "independent variable" (line 287) and "dependent variable" (line 288) should be replaced with "predictor variable" and "criterion variable," respectively. Even a longitudinal design (as suggested on lines 432 to 434) would not address the issue of causation, although confidence in the model might increase.

3. Given the relatively minimal reductions in beta values in the mediation analyses, the findings should be described as reflecting only partial mediation (as they are in the Abstract). With partial mediation, the statement on lines 359 and 360 is not true.

4. It is plausible that the three sets of variables (i.e.,life stress, negative thoughts, burnout) could be causally related in multiple ways. For example, life stress could precipitate burnout, which could lead to negative thoughts. It would be of interest to see how other potential models pan out.

Additional comments

1. The applied implications, which begin on line 440, are not necessarily wrong, but they seem premature in light of the correlational nature of the results.

Reviewer 2 ·

Basic reporting

This study tried to identify the stress-burnout relationship and the meditating role of negative thoughts to explain such relationship in the Taiwanese student athletes. Such investigation seems to be meaningful to support cognitive-affective model of athletic burnout which was developed in the western society and verify validation of the model in order to apply it to the different country which has different cultural norm. Moreover, this study is aiming at test meditating effect of negative thoughts between stress and burnout in the athletic context and it is strength of the current study.

However, there are many points that should be addressed and corrected throughout the entire manuscript. More details are indicated at below and the authors should revise them.

* In the entire manuscript, many grammatic errors, awkward sentences, and illogical paragraphs have been found and indicated in the pdf manuscript by the reviewer. Therefore, it is highly suggested that the manuscript should be edited by English professionals before resubmitted.

1. Introduction

-It is too long and very confused to catch up what the authors want to describe.
For instance, negative thoughts, just only one research variable in this study was described in about 3 pages from page 14-17. It needs to be concisely shorten.
-Moreover, the necessity of the study was not enough to support this study. It also needs to add some more sentences.
- The purpose of the study should be clearly indicated.

2. Methods and Results

These sections have been relatively logically written. Results section in particular is well described the findings. However, some awkward sentences have been found.

3. Discussion

It has been relatively well described. However, it is also too long, almost 10 pages. The authors should concisely rewrite.

4. References

It is 15 pages and therefore should be shorten (suggested less than 40 references).

Experimental design

2. Methods and Results

These sections have been relatively logically written. Results section in particular is well described the findings. However, some awkward sentences have been found.

Validity of the findings

2. Methods and Results

These sections have been relatively logically written. Results section in particular is well described the findings. However, some awkward sentences have been found.

Additional comments

This study is timely appropriate to identify the stress-burnout relationship and the mediating role of negative thoughts for the Taiwanese athletes. In addition, the findings will contribute to develop psychological knowledge and athletic performance in Taiwan. However, there are many aspects that should be corrected or addressed by the authors in entire manuscript.

Therefore, it is not enough to be considered in publication without major revision. I suggest that this manuscript should be resubmitted after revision.

Annotated reviews are not available for download in order to protect the identity of reviewers who chose to remain anonymous.

---

## Round 0.2 · Minor Revisions

I now have received two reviewers' comments on your revised manuscript. Although both reviewers are generally satisfied with your reply and revision based on previous comments, one reviewer has pointed out some issues that require your additional attention. Please address these issues and provide a point by point reply in addition to the revised manuscript. I look forward to receive your revised manuscript.

Tsung-Min Hung, Ph.D.
PeerJ editor
Distinguished professor
Department of Physical Education
National Taiwan Normal University

Reviewer 1 ·

Basic reporting

I appreciate the efforts of the authors in addressing my concerns and those of the other reviewer. The manuscript is very much improved as a result of the changes that have been made.

Experimental design

In my previous review, I should have noted that evidence in support of the validity of the CSALSS, ATQ, and ABQ should be summarized in the Measures section (not the Results section, as I mistakenly wrote). What support for the validity of the scales, independent of the results of the current study, has been obtained?

Validity of the findings

Although I appreciate inclusion of the sentences on lines 439-445 about examining other models with the three sets of variables, it would be ideal if the authors could report some of those analyses in the current paper since they have the data to do so. Findings from such analyses could strengthen confidence in the main cross-sectional findings, particularly if other models don't work as well as the one tested in the main analyses.

Reviewer 2 ·

Basic reporting

It is witnessed that the manuscript throughtout the entire texts has been clearly improved.

Experimental design

These sections have been relatively logically written. Results section in particular is well described the findings.

Validity of the findings

The findings will contribute to develop psychological knowledge and athletic performance in Taiwan.

Additional comments

It has been enoughly improved throughout the entire manuscript and so I believe that the manuscript is OK to be considered in publication without revision.

---

## Round 0.3 · accepted · Accept

I have read through your reply to the reviewer's comment and your revised manuscript. I am satisfied with your response. You and your coauthors have my congratulations. Thank you for choosing PeerJ as a venue for publishing your research work and I look forward to receiving more of your work in the future.

Tsung-Min Hung, Ph.D.
PeerJ editor
Distinguished professor
Department of Physical Education
National Taiwan Normal University